# Downregulation of mitochondrial biogenesis by virus infection triggers antiviral responses by cyclic GMP-AMP synthase

Hiroki Sato[1,2], Miho Hoshi[3], Fusako Ikeda[4], Tomoko Fujiyuki[1,5], Misako Yoneda[4], Chieko Kai[1] *

1 Infectious Disease Control Science, Institute of Industrial Science, The University of Tokyo, Tokyo, Japan, 2 Molecular Virology, Institute of Industrial Science, The University of Tokyo, Tokyo, Japan, 3 The Institute of Medical Science, The University of Tokyo, Tokyo, Japan, 4 Division of Virological Medicine, Institute of Industrial Science, The University of Tokyo, Tokyo, Japan, 5 Virus Engineering, Institute of Industrial Science, The University of Tokyo, Tokyo, Japan

* ckai@iis.u-tokyo.ac.jp

**Data Availability Statement:** Microarray data of measles virus are available from https://ddbj.nig.ac.jp/public/ddbj_database/gea/experiment/E-GEAD-000/E-GEAD-331/ Microarray data of other viruses

## Abstract

In general, in mammalian cells, cytosolic DNA viruses are sensed by cyclic GMP-AMP synthase (cGAS), and RNA viruses are recognized by retinoic acid-inducible gene I (RIG-I)-like receptors, triggering a series of downstream innate antiviral signaling steps in the host. We previously reported that measles virus (MeV), which possesses an RNA genome, induces rapid antiviral responses, followed by comprehensive downregulation of host gene expression in epithelial cells. Interestingly, gene ontology analysis indicated that genes encoding mitochondrial proteins are enriched among the list of downregulated genes. To evaluate mitochondrial stress after MeV infection, we first observed the mitochondrial morphology of infected cells and found that significantly elongated mitochondrial networks with a hyperfused phenotype were formed. In addition, an increased amount of mitochondrial DNA (mtDNA) in the cytosol was detected during progression of infection. Based on these results, we show that cytosolic mtDNA released from hyperfused mitochondria during MeV infection is captured by cGAS and causes consequent priming of the DNA sensing pathway in addition to canonical RNA sensing. We also ascertained the contribution of cGAS to the in vivo pathogenicity of MeV. In addition, we found that other viruses that induce downregulation of mitochondrial biogenesis as seen for MeV cause similar mitochondrial hyperfusion and cytosolic mtDNA-priming antiviral responses. These findings indicate that the mtDNA-activated cGAS pathway is critical for full innate control of certain viruses, including RNA viruses that cause mitochondrial stress.

## Author summary

Viruses exert their pathogenicity by targeting various cellular components in infected cells. In response, host cells have evolved strategies to sense intracellular pathogen-associated molecules, such as nucleic acids derived from infected virus, and trigger subsequent

are available from GEO (https://www.ncbi.nlm.nih.gov/geo/), accession numbers: GSE32140 for RSV, GSE38866 for VSV, GSE53103 for SeV and GSE72397 for CPV.

**Funding:** This work was supported by the grant from the Japan Society for the Promotion of Science (JSPS) KAKENHI (No. JP16H02587) awarded to C.K. The funders had no role in study design, data collection and analysis, decision to publish, or preparation of the manuscript.

**Competing interests:** The authors have declared that no competing interests exist.

antiviral responses to counteract infection. Measles virus (MeV), the causative agent of human measles, is the most highly contagious virus, killing 300 children per day worldwide; thus MeV has been targeted for eradication by the World Health Organization. In the present study, we found that MeV causes downregulation of mitochondrial biogenesis accompanied with aberrant hyperfusion of mitochondria in the infected cells. Furthermore, we show that cytoplasmic release of mitochondrial DNA activates DNA sensor molecule, cGAS, in addition to the innate immune response induced by the viral component. Importantly, this phenomenon was also observed for viruses, both RNA and DNA, which target mitochondrial biogenesis. Our study provides new insights into the mitochondrial stress by virus infection and an important host defense system to suppress viral propagation.

## Introduction

Innate immunity provides the first line of defense in the host against invading microbes [1], utilizing a series of pattern recognition receptors (PRRs) to recognize pathogen-associated molecular patterns (PAMPs) that are present on microbes. The Toll-like receptor (TLR) family of proteins, which are expressed on innate immune cells such as dendritic cells, macrophages, and neutrophils, detect extracellular PAMPs. Microbes can also deliver PAMPs to the cytosol of host cells, which are surveyed by intracellular PRRs [2]. A heterogeneous group of PRRs detect nucleic acids; these include the RNA sensors retinoic-acid-inducible gene I (RIG-I) and melanoma differentiation-associated protein 5 (MDA5), and the DNA sensor cyclic GMP-AMP synthase (cGAS). RNA viruses activate RIG-I and MDA5, followed by stimulation of signaling via mitochondrial antiviral-signaling protein (MAVS). DNA viruses activate cGAS, which produces 2′3′ cGAMP to stimulate stimulator of interferon genes (STING). Both types of signaling lead to the activation of the TBK1-IRF3-IFN-β pathway [1,3]. This process of nucleic acid detection is tightly regulated because of host nucleic acids inside cells.

Measles virus (MeV), which possesses a single-stranded negative-sense RNA genome (-ssRNA), is one of the most important pathogens in humans, and a major cause of child mortality, particularly in developing countries [4], and has been targeted for eradication by the World Health Organization [5]. MeV infection causes several characteristic syndromes, such as fever, rash, immunosuppression, and life-long immunity. As with other RNA viruses, MeV infection triggers the RNA sensing pathway [6,7], inducing the rapid activation of the innate antiviral response [8–10] and the consequent production of various cytokines [9,11,12] in epithelial cells. Furthermore, our previous study using microarray analysis revealed that many genes, including antiviral factors, are upregulated [13]. Interestingly, downregulation of numerous genes, especially housekeeping genes, has also detected during the late stage of infection [14].

In this study, we found that MeV infection induces aberration of mitochondrial morphology with hyperfusion and subsequent liberation of mitochondrial DNA (mtDNA) into cytoplasm, which leads to activation of the cGAS cascade in addition to the actual RNA sensing pathway. We further show that this cascade is common among viruses, both DNA and RNA, causing the downregulation of mitochondrial biogenesis. These findings reveal a novel host defense strategy for suppressing viral propagation.

## Results

### Measles virus infection results in hyperfusion of mitochondria

Our previous study demonstrated that MeV induces characteristic comprehensive downregulation of housekeeping genes in epithelial cells [13,14] (Data set: [15]). We first assigned each downregulated gene to a cellular component group. Interestingly, gene ontology analysis revealed significant overrepresentation of cellular components related to mitochondria (Figs 1A and S1A). To confirm whether MeV infection actually causes downregulation of mitochondrial biogenesis, the mitochondrial mass was measured and a decrease in the mitochondrial mass was observed in the MeV-infected cells (S1B Fig). The mitochondrial membrane potential was also lowered by MeV infection to the same extent as treatment with depolarizing reagent (S1C Fig). In addition, the total amount of mtDNA was decreased (S1D Fig). Taken together, these findings indicate that downregulation of mitochondrial biogenesis after MeV infection is substantial.

Previous studies have demonstrated that cellular stress conditions induce a transient change in the highly fused network morphology of mitochondria, which is considered an adaptive process [16,17]. To assess morphological changes in mitochondria after MeV infection, we observed MeV-infected cells. Confocal microscopy of cells after MeV infection revealed significantly elongated, interconnected mitochondrial networks consistent with a hyperfused phenotype (Fig 1B and 1C). In particular, multinuclear giant cells resulting from MeV infection, which were indicated by EGFP produced by recombinant MeV (rMV-EGFP) [18], showed aberrant hyperfusion of mitochondria accompanied by the formation of a mesh of highly interconnected, thin mitochondrial filaments, compared with contiguous uninfected cells in the same microscopic field (Fig 1B). We further examined the implications of syncytia formation on mitochondrial hyperfusion with coexpression of the MeV-F and H proteins, and we found that the mitochondrial morphology was remained unchanged (S1E Fig). Other cell lines susceptible to MeV also showed a hyperfused phenotype in the mitochondria of infected cells (S2A Fig). To observe dynamic morphological changes after MeV infection, rMV-EGFP-infected cells were analyzed by time-lapse fluorescence microscopy. The mitochondrial shape in multinuclear giant cells was first entangled, and then formed a hyperfused phenotype accompanied by the progression of virus replication (S1–S3 Movies). To reveal whether the morphological changes were a consequence of mitophagy, which is caused by mitochondrial damage, intracellular localization of EGFP-fused LC3 was observed as a marker of autophagy. After MeV infection, LC3 formed dots in the cytoplasm, as described previously [19–21], while colocalization with mitochondria was not observed (S2B Fig). These findings suggested that the morphological changes were not the result of mitochondrial damage.

### mtDNA released into the cytosol by MeV infection is captured by cGAS and induces host antiviral responses

Recent reports have demonstrated that several types of mitochondrial stress cause the release of cytosolic mtDNA from mitochondria, which can trigger antiviral responses [22,23]. In particular, mtDNA stress induced by herpes simplex virus 1 (HSV-1), a DNA virus, infection results in both hyperfusion of mitochondria and release of mtDNA into the cytosol, and the consequent priming of innate immune responses via the cytosolic DNA sensor cGAS [24]. We assayed for extramitochondrial DNA after MeV infection. Analysis of pure cytosolic extracts revealed an increase in cytosolic mtDNA over the time course of MeV infection (Figs 2A and S3A). To confirm that cytosolic mtDNA activates cGAS via a direct interaction, we immunoprecipitated cGAS from MeV-infected cells and utilized qPCR analysis to detect coprecipitated

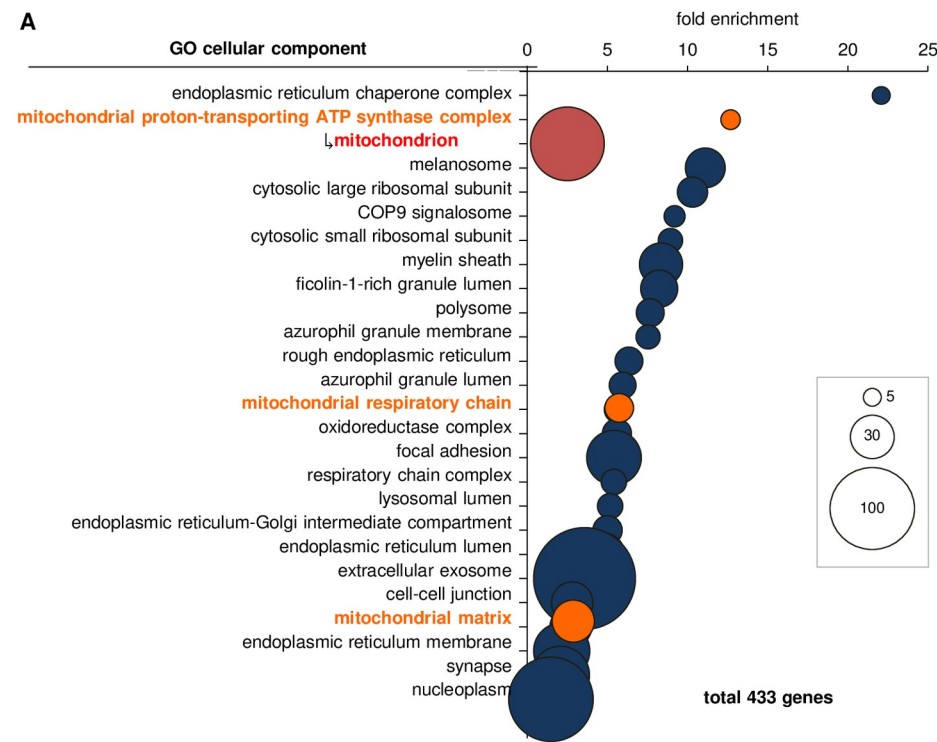

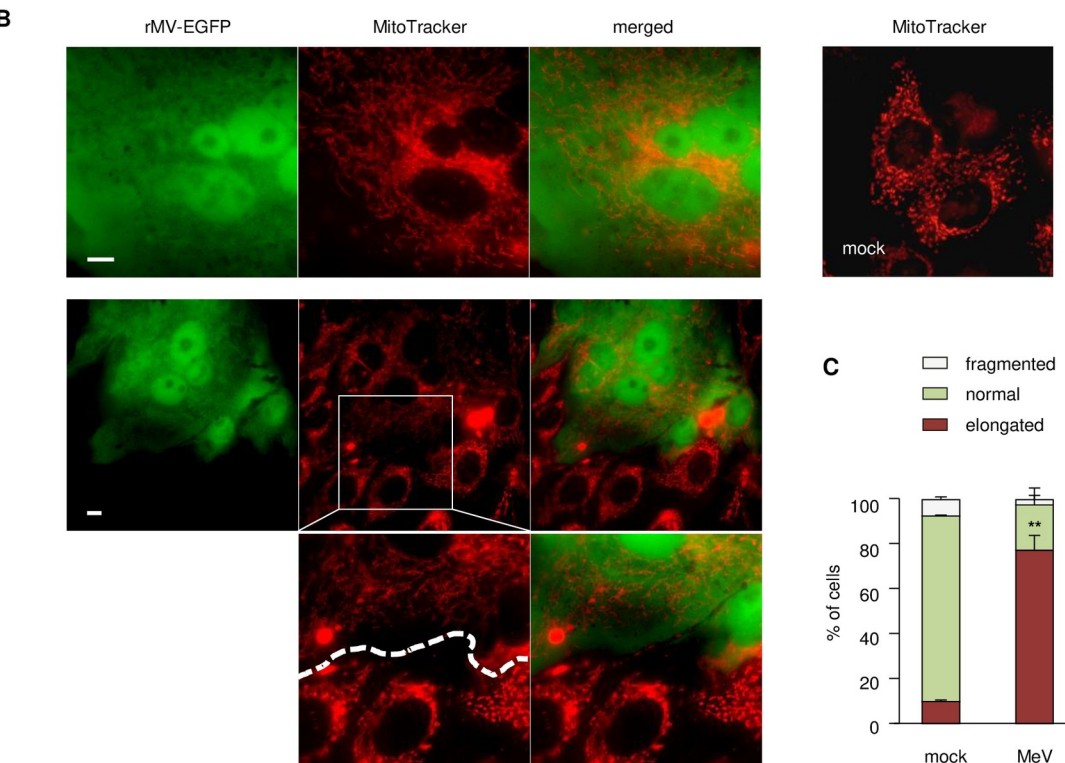

**Fig 1. MeV infection induces downregulation of mitochondrial biogenesis and hyperfusion of mitochondria.** (A) Gene ontology analysis of the cellular components overrepresented among genes downregulated by MeV infection. Uppermost layers in the list are shown on the vertical axis, and respective enrichment scores are shown on the horizontal axis. Each gene number is represented by the size of the circle. Mitochondria-related components are shown in orange. Mitochondrion, which

is the bottommost and largest term in the mitochondria-related hierarchy, is shown in red below the highest items of the mitochondria-related components. (B) Confocal microscopy images of Vero-hSLAM cells infected with rMV-EGFP after 16 h. Mitochondria were stained with MitoTracker. Scale bar = 10 μm. Lower images are enlargements of squared region. (C) Quantification of immunofluorescence shown in (B). Mitochondrial networks of > 40 cells per condition and experiment ($n = 3$) were classified into three morphological categories; normal, elongated, and fragmented. Data are the mean value ± SD ($n = 3$). Statistical significance was determined using unpaired Student's $t$-test; $**P < 0.01$.

mtDNA. Although there was no mtDNA enrichment when cGAS was immunoprecipitated from mock-treated cells, a significant enrichment of mtDNA was observed in cells infected with MeV (Fig 2B). Under the same conditions, genomic DNA was under the detectable level (Fig 2B). We next performed knockdown of sensor molecules for intracellular foreign nucleic acids. As expected, knockdown of MAVS, immediately downstream of RNA sensor molecules, decreased interferon (IFN)-β production following MeV infection, while cGAS knockdown also suppressed it to an extent (Fig 2C). The induction of other antiviral factors after MeV infection also decreased as a result of cGAS depletion (S3B Fig). We observed the mitochondrial morphology after MeV infection in MAVS knockdown cells, and found that mitochondrial hyperfusion was the same as in normal control cells (S3C Fig). This indicated that the formation of hyperfused mitochondria is a distinct process from that of the assembly of activated MAVS on the outer mitochondrial membrane induced by RNA virus infections [25]. On the other hand, vesicular stomatitis virus (VSV), which possesses a -ssRNA genome like MeV, showed no effect on IFN-β production following cGAS knockdown (Fig 2C) as previously reported [26,27]. We next confirmed the kinetics of activation of RNA/DNA sensors after MeV infection. A luciferase assay driven by type-I interferon stimulated response elements (ISRE) was measured following MeV infection over a set time course. As expected, rapid activation of ISRE which might be induced by a typical RNA sensor against MeV infection was detected (Fig 2D). Furthermore, phosphorylation of STING which is a hallmark of cGAS activation was detected from 1 dpi, when ISRE activation was diminishing (Fig 2D). To prove the direct involvement of mtDNA release in antiviral responses in MeV-infected cells, we used dideoxycytidine (ddC), a deoxyribonucleoside analogue that specifically inhibits mtDNA replication [28,29]. Treatment of cells with ddC resulted in a reduced mtDNA copy number (Fig 2E). ddC treatment did not influence on the cell viability and MeV growth (S3D Fig), while diminished IFN-β production by MeV (Fig 2F) to a similar extent as cGAS knockdown (Fig 2C). By contrast, VSV was not affected by ddC (Fig 2F). To further confirm the implication of cGAS after MeV infection, we used 293SLAM cells [13], which were established based on HEK293 cells, in which expression of cGAS was negligible [30] (Fig 2G). mtDNA depletion in 293SLAM cells did not alter IFN-β expression following MeV infection (Fig 2G). Furthermore, transient expression of cGAS in 293SLAM cells resulted in a higher level of IFN-β induction after MeV infection (Fig 2H).

These data indicated that MeV infection promotes the liberation of mtDNA from mitochondria, which accesses the cytosol to engage in innate immune signaling triggered by cGAS.

## Downregulation of mitochondrial biogenesis results in priming of antiviral responses

We next investigated how MeV infection induces IFN-β expression via a cGAS-dependent pathway. To confirm whether the cytosolic release of mtDNA is a consequent of mitochondrial hyperfusion, we first tested artificial defects of mitochondrial hyperfusion by knockdown of mitofusin 1 (Mfn1) (Fig 3A), which plays a key role in the fusion of mitochondria. Mfn1 depletion induced no apparent hyperfusion but rather fission of mitochondria in MeV-infected cells (Fig 3B and 3C). Furthermore, the increase in the amount of cytosolic mtDNA in MeV-

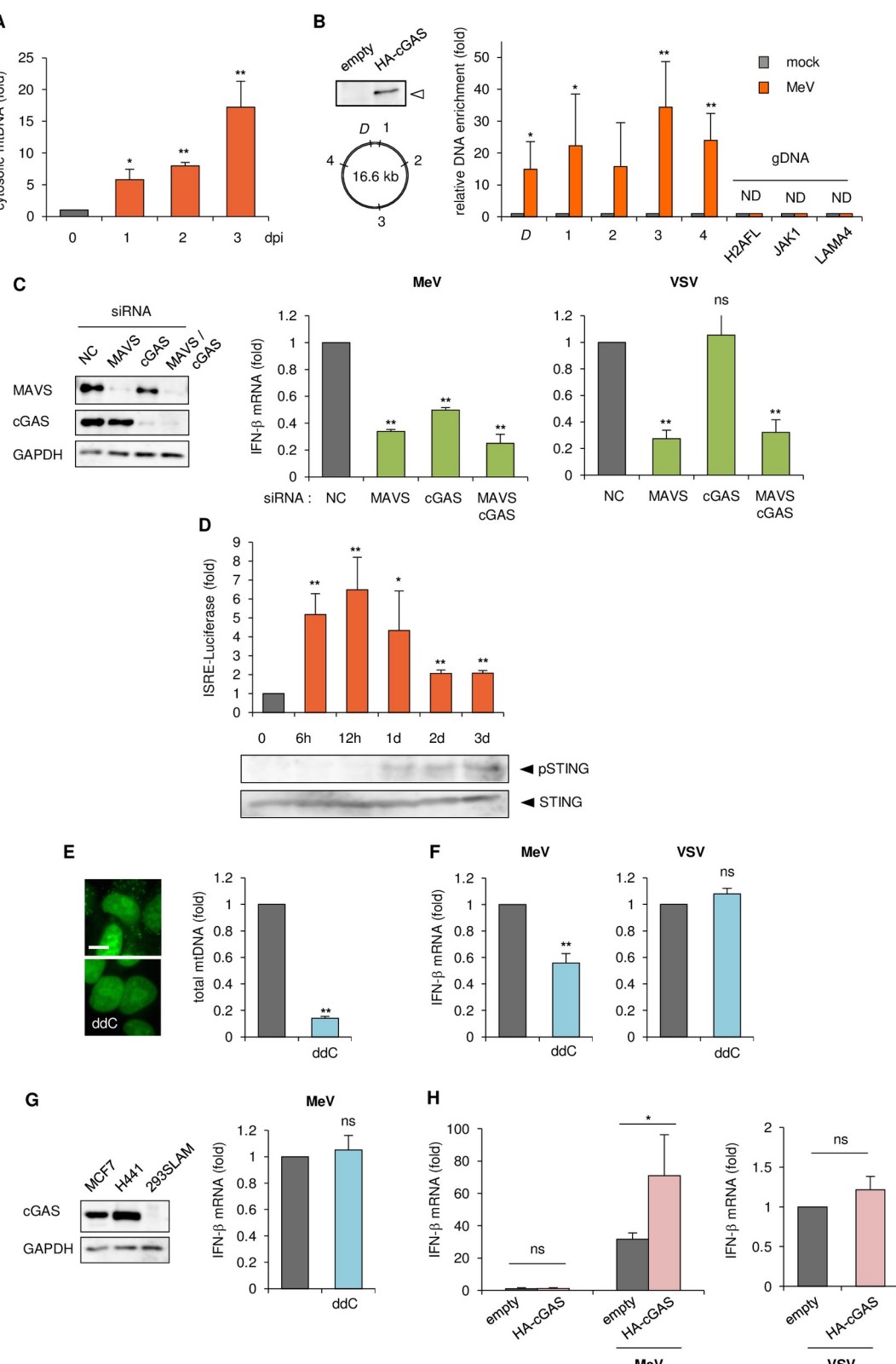

**Fig 2. The role of cGAS in innate antiviral responses against MeV.** (A) Levels of mtDNA present in the cytosol of MeV-infected MCF7 cells over time. Cytosolic mtDNA was quantitated via qPCR using a mitochondrial *Dloop* primer set (*D*), and represented as fold increase relative to mock-treated cells. (B) Immunoprecipitation followed by qPCR. Upper left:

immunoblot of cell lysate transfected with empty plasmid or plasmid expressing HA-tagged cGAS. Lower left: the relative location of qPCR primer sets on mtDNA. Right: enrichment of DNA fragments using anti-HA antibody to coprecipitate DNA in mock- or MeV-infected cells, represented as fold increase relative to mock-treated cells. DNA fragments were amplified by qPCR using five primer pairs for mtDNA and three primer pairs for genomic DNA (gDNA). (C) MCF7 cells were transfected with siRNA for the negative control (NC), MAVS, cGAS, or both. Left: immunoblot of the cell lysate. Right: cells were infected with MeV or VSV, and the RNA collected at 24 h post-infection (hpi) was analyzed for IFN-β expression by RT-qPCR. (D) MCF7 cells were co-transfected with pISRE-Luc which is induced by type I IFN, and phRL-TK(int-) as an internal control, and then infected with MeV. Upper: Cells were harvested at the indicated time and the luciferase activities were measured. Lower: Cell lysates were subjected to western blotting to detect endogenous STING and phosphorylation of STING caused by cGAS activation. (E) qPCR analysis of the mtDNA content of cells cultured in ddC to generate mtDNA-depleted cells. Representative image of MCF7 cells stained with PicoGreen nucleic acid stain. Scale bar = 10 μm. (F) MCF7 cells treated with or without ddC were infected with MeV or VSV, and the RNA collected at 24 hpi was subjected to RT-qPCR to analyze IFN-β expression. (G) Left: cGAS expression levels in MCF7, H441, and 293SLAM cells were confirmed by western blotting. Right: 293SLAM cells treated with or without ddC were infected with MeV, and IFN-β mRNA levels were measured by RT-qPCR. (H) 293SLAM cells were transfected with empty plasmid or plasmid expressing HA-cGAS, and then infected with MeV or VSV. RNA was collected at 24 hpi and IFN-β levels were measured by RT-qPCR. Data are representative of three independent experiments. Data are the mean value ± SD ($n$ = 3). Statistical significance was determined using one-way analysis of variance (ANOVA) followed by Dunnett's multiple comparison test (A, C, D) or unpaired Student's $t$-test (B, E–H); $^*P < 0.05$; $^{**}P < 0.01$; ns, not significant ($P > 0.05$).

infected cells was significantly reduced by Mfn1 knockdown (Figs 3D and S4A). These results showed that mitochondrial hyperfusion and the liberation of mtDNA by MeV infection is conducted by Mfn1. Previous reports indicated that cellular stress induced by treatment with actinomycin D (ActD) or UV irradiation promoted mitochondrial hyperfusion, which is also mediated by Mfn1 [17]. We also confirmed that the ActD or UV treatment, or overexpression of Mfn1 caused mitochondrial elongation (Fig 3E and 3F), which was induced by Mfn1 (S4B Fig). Interestingly, these treatments caused neither liberation of mtDNA nor upregulation of IFN-β expression (Fig 3G). indicating that the process of canonical fusion of mitochondria sequesters mtDNA in mitochondria. These findings suggested that the mitochondrial fusion involving Mfn1 is required for mtDNA release, but additional process(es) contribute to mtDNA liberation induced by MeV infection.

We next analyzed whether the downregulation of mitochondrial biogenesis characteristically observed in MeV-infected cells is implicated in mtDNA release. Peroxisome proliferator-activated receptor gamma-coactivator-1α (PGC-1α) is known to bind to and consequently modulate the activity of several transcription factors [31–35]. In particular, PGC-1α is a co-regulator of nuclear respiratory factor (NRF) 1 and 2, which govern the expression of nuclear encoding factors involved in mitochondrial transcription, the mitochondrial protein import machinery and mitochondrial translation factors [36–39]. After MeV infection, the expression levels of PGC-1α and TFAM (transcription factor A, mitochondrial), which contribute to mitochondrial biogenesis via mtDNA transcription, were not altered (S4C Fig). Thus, to mimic the transcriptional downregulation of mitochondrial proteins artificially, knockdown of PGC-1α was performed (Fig 3H). Nuclear genes encoding mitochondrial proteins were decreased in expression by PGC-1α knockdown, but no alterations or upregulation of non-mitochondrial protein genes were observed (S4D Fig upper panel). The mitochondrial membrane potential was also decreased by PGC-1α knockdown (S4E Fig). Under these conditions, cytosolic mtDNA was increased (Figs 3I and S4F), and simultaneous upregulation of IFN-β and IFN-stimulated genes (ISGs) was detected (Figs 3J and S4D lower panel). As expected, mtDNA depletion by ddC rendered a decrease in IFN-β upregulation by PGC-1α knockdown (Fig 3J). Taken together, these findings suggested that the comprehensive downregulation of mitochondrial biogenesis causes cytosolic release of mtDNA and consequent antiviral priming.

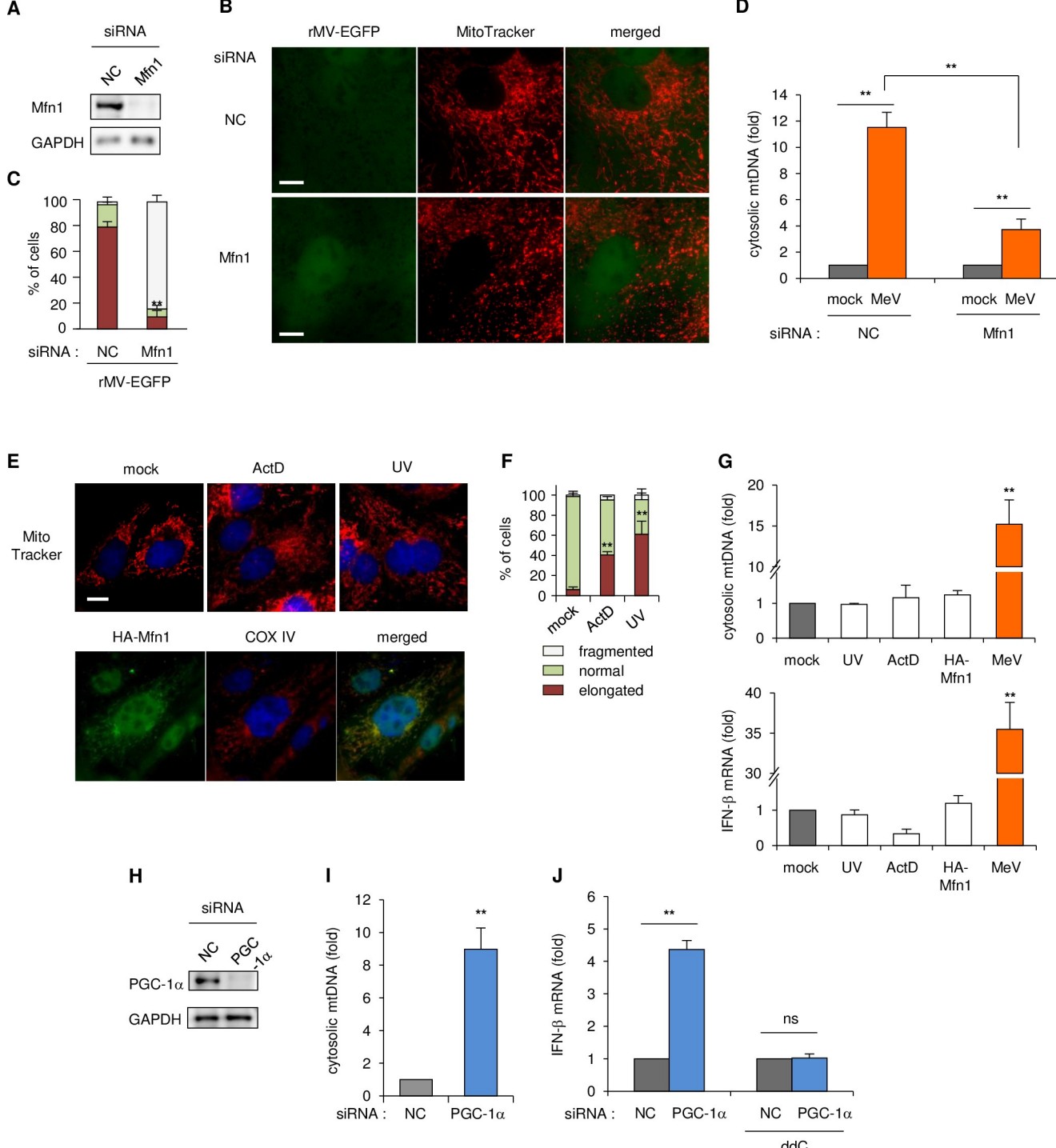

**Fig 3. The role of Mfn1 in the downregulation of mitochondrial biogenesis in the mtDNA-priming antiviral response.** (A-C) Vero-hSLAM cells were transfected with siRNA for the NC or Mfn1. (A) Cell lysates analyzed to western blotting. (B) Cells infected with rMV-EGFP. At 16 hpi, mitochondria were stained with MitoTracker. Scale bar = 10 μm. (C) Mitochondria morphology of at least 30 cells per condition and in three independent experiments were classified as normal, elongated, or fragmented mitochondrial network. (D) Mfn1 knockdown cells were infected with the mock control or MeV, and cytosolic mtDNA levels were measured by qPCR, as described above. (E) Upper: Vero-hSLAM cells were treated with 3 μg/ml ActD or irradiated with 60 mJ/cm$^2$ UV-C. Mitochondria and nuclei were stained 7 h later with MitoTracker and Hoechst, respectively. Lower: Cells were transfected with plasmid expressing HA-tagged Mfn1 and then stained with antibodies to COX IV for mitochondria and to the HA tag, and Hoechst. Scale bar = 10 μm. (F) Mitochondrial morphology of ~30 cells per condition and in two experiments were classified as normal, elongated, or fragmented mitochondrial network. (G) Cells were irradiated with UV, treated with ActD for 7 h, transfected with HA-Mfn1 plasmid, or infected with MeV for 24 h. Upper: cells were harvested and cytosolic

mtDNA was measured by qPCR, as described above. Lower: total RNA was subjected to RT-qPCR to analyze IFN-β mRNA. (H) Immunoblot of MCF7 cells transfected with siRNA for the NC or PGC-1α. (I) MCF cells transfected with siRNA for the NC or PGC-1α, and cytosolic mtDNA was quantified by qPCR at 5 d post-transfection. Data are represented as the relative number of PGC-1α knockdown cells to that of NC-transfected cells. (J) MCF7 cells were treated with the mock control or ddC for 3 d and then transfected with siRNA for the NC or PGC-1α. After 4 d, RNA was harvested and the mRNA levels of IFN-β were measured by RT-qPCR. Data are representative of three independent experiments. Data are the mean value ± SD ($n = 3$). Statistical significance was determined using an unpaired Student's $t$-test; $^*P < 0.05$; $^{**}P < 0.01$; ns, not significant ($P > 0.05$).

## Contribution of cGAS to the *in vivo* pathogenicity of MeV

To further confirm the implication of cGAS in antiviral responses during MeV infection *in vivo*, we used a rodent brain-adapted strain of MeV (MeV-CAMR40) [40,41]. Knockout mice lacking the RNA sensor-related molecule (MAVS) or DNA sensor molecule (cGAS) were intracerebrally inoculated with the virus, and the survival curve was measured. As expected, knockout mice of MAVS ($MAVS^{-/-}$) were more vulnerable to MeV infection compared with wild-type mice (Fig 4A). Importantly, $cGAS^{-/-}$ mice also showed severe symptoms and the mortality rate was close to that of $MAVS^{-/-}$ mice (Fig 4A). MeV replication measured by MeV-N expression in the brains of $cGAS^{-/-}$ mice was higher than that in wild-type mice (Fig 4B). A decrease in the relative amount of IFN-β production to MeV replication was observed in $cGAS^{-/-}$ mice (Fig 4C). Similarly, the expression of other ISGs was also suppressed in $cGAS^{-/-}$ mice (S5 Fig). These results indicated that cGAS plays an important role in the antiviral response in MeV infection *in vivo* as well as MAVS.

## A role for mtDNA release in innate antiviral responses against RNA and DNA viruses

We investigated whether other viruses also induce downregulation of mitochondrial biogenesis. We searched microarray databases or previous reports of related viruses that cause downregulation of host gene expression, in which the list of downregulated genes included enriched mitochondrial protein genes. Among -ssRNA viruses, respiratory syncytial virus (RSV) shows significant downregulation of genes encoding mitochondrial proteins after infection (the fold-enrichment of mitochondria-related genes was low, but the total number of genes was high)

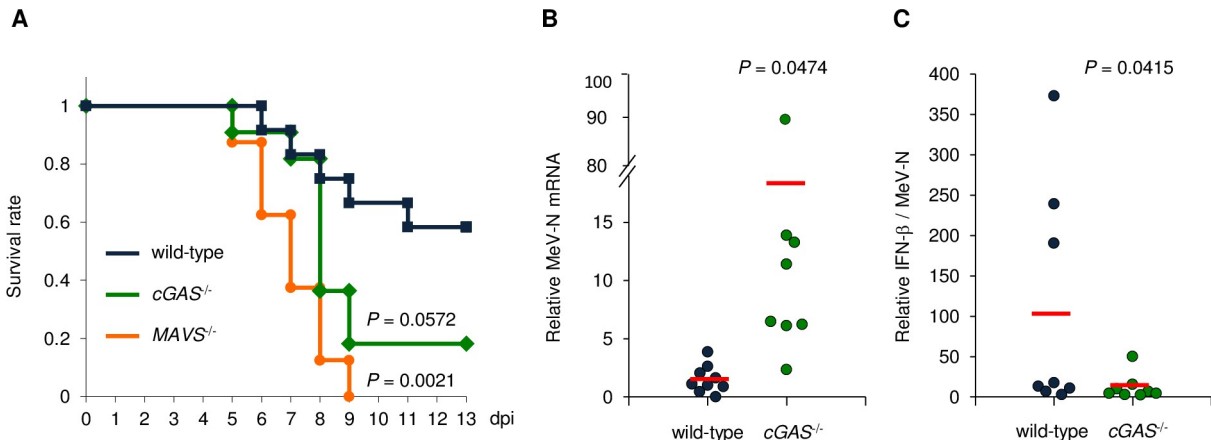

**Fig 4. Requirement for cGAS in controlling viral infection in vivo.** (A) Survival curves of wild-type ($n = 13$, male = 5, female = 8), $MAVS^{-/-}$ ($n = 8$, male = 4, female = 4), and $cGAS^{-/-}$ ($n = 11$, male = 8, female = 3) mice intracerebrally infected with $1.0 \times 10^3$ TCID$_{50}$ of MeV-CAMR. (B, C) wild type or $cGAS^{-/-}$ mice ($n = 8$, male = 3, female = 5) were intracerebrally inoculated with $1.0 \times 10^3$ TCID$_{50}$ of MeV-CAMR, and the cerebrum was harvested at 5 dpi. (B) mRNA of the MeV-N gene measured by RT-qPCR. (C) IFN-β measured by RT-qPCR, represented as amount measured relative to the amount of MeV-N. Data are representative of three independent experiments. Data are the mean value ± SD ($n = 3$). Statistical significance was determined by the log rank test (A) or unpaired Student's $t$-test (B, C).

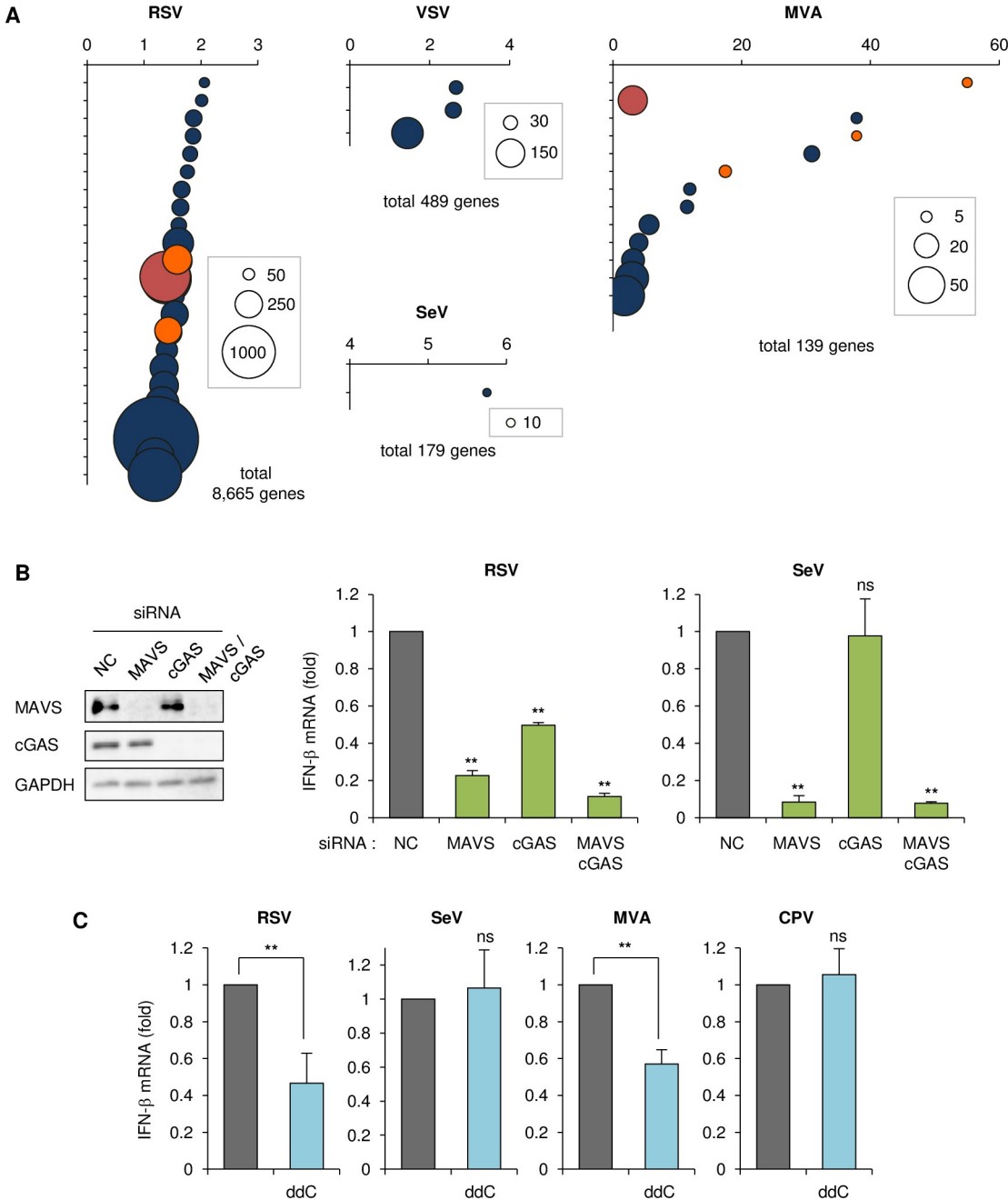

**Fig 5. RNA and DNA viruses that cause downregulation of mitochondrial biogenesis induce mtDNA-priming IFN-β production.** (A) Gene ontology analysis of the cellular components overrepresented among genes downregulated by RSV, VSV, SeV, and MVA infection. Each item in the graph is represented as shown for Fig 1A. Uppermost layers in the list are shown in S6 Fig CPV showed no obvious enrichment in cellular components. (B) Left: HEp-2 cells were transfected with siRNA for the NC, MAVS, cGAS, or both, followed by western blotting. Right: Cells were infected with RSV or SeV, and the RNA collected at 24 hpi was analyzed for IFN-β by RT-qPCR. (C) Cells treated with or without ddC were infected with RSV, SeV, MVA or CPV, and the RNA collected at 24 hpi was analyzed for IFN-β by RT-qPCR. Data are representative of three independent experiments. Data are the mean value ± SD (n = 3). Statistical significance was determined using one-way ANOVA followed by Dunnett's multiple comparison test (B), or unpaired Student's t-test (C); $^*P < 0.05$; $^{**}P < 0.01$; ns, not significant ($P > 0.05$).

(Figs 5A and S6) [42]. While other -ssRNA viruses, including VSV [43] and Sendai virus (SeV) [44], showed no obvious enrichment in mitochondria-related genes (Figs 5A and S6). Among DNA viruses, modified vaccinia virus Ankara strain (MVA) displayed marked enrichment of mitochondrial proteins (Figs 5A and S6) [45]. We included canine parvovirus (CPV) in our analysis for comparison, which induces the downregulation of ~300 genes, but no apparent enrichment in any cellular component including mitochondria was detected [46,47]. We first observed mitochondrial morphology after infection. RSV-infected cells showed marked elongation of mitochondria, similar to that seen with MeV, whereas VSV and SeV induced no alterations in mitochondrial morphology (Fig 6A and 6B). Similarly, MVA but not CPV induced elongation of mitochondria (Fig 6A and 6B). We further confirmed the implication of cGAS on IFN-β production by RNA viruses. SeV infection was affected by MAVS knockdown but not cGAS knockdown, as previously reported [26], whereas IFN-β expression following RSV infection was suppressed by cGAS knockdown, similar to MeV infection (Fig 5B). Importantly, ddC treatment diminished IFN-β expression after infection with RSV and MVA, but not other viruses (Fig 5C), which correlated with the downregulation of mitochondrial proteins and the formation of hyperfused mitochondria.

From these data, we propose that viruses which possess the potential to intrinsically downregulate mitochondrial biogenesis, as seen for MeV, activate cGAS-dependent antiviral responses via the liberation of mtDNA to the cytosol by the hyperfusion of mitochondria. This cascade is considered to be required for full innate antiviral responses against these viruses.

## Discussion

Previous studies have demonstrated that the MeV RNA genome is recognized by RIG-I and MDA5 [6,7], which in turn activates MAVS inducing the innate RNA sensing pathway, as seen for other -ssRNA viruses.

A key finding of the present study is that MeV stimulates type I IFN and ISG expression in a cGAS-dependent manner, which is known as the canonical sensing pathway caused by DNA virus infection. In MeV infection, we propose a two-step induction of antiviral responses; at an early phase of infection, viral RNA replication is detected rapidly by an RNA sensor, while during the late phase of infection, mitochondrial downregulation accompanied by mtDNA liberation causes prolonged IFN-β and ISG production (Fig 2D). These findings uncover a novel host strategy of the defense system for suppressing viral propagation.

Under normal circumstances, the cytoplasm is devoid of DNA. Nevertheless, several recent reports have revealed that mtDNA can gain access to the cytoplasm under certain circumstances of stress or damage, and provoke at least three pathways for innate immune responses; mtDNA acts as a damage-associated molecular pattern in inflammation initiation through direct activation of TLR9, which usually recognizes bacterial DNA [48–50]. In addition, mtDNA released into the cytoplasm also plays a key role in activation of the NLRP3 inflammasome and mediates the secretion of IL-1β and IL-18 [51,52]. Furthermore, as described above, degradation of mtDNA, termed mtDNA stress, induced by HSV infection or depletion of mtDNA-binding protein TFAM leads to cytoplasmic release of mtDNA and the consequent activation of the cGAS-STING pathway [24]. Therefore, in addition to its well-appreciated roles in cellular metabolism and energy production, mtDNA can be identified as an intrinsic cellular trigger of antiviral signaling and cellular monitoring of mtDNA homeostasis cooperates with established virus sensing mechanisms.

West and colleagues revealed that HSV-1 infection causes mtDNA stress, and induces mitochondrial hyperfusion conducted by Mfn1 [24], but the detail of this process remains to be elucidated. In the present study, we revealed that MeV also induced mitochondrial hyperfusion

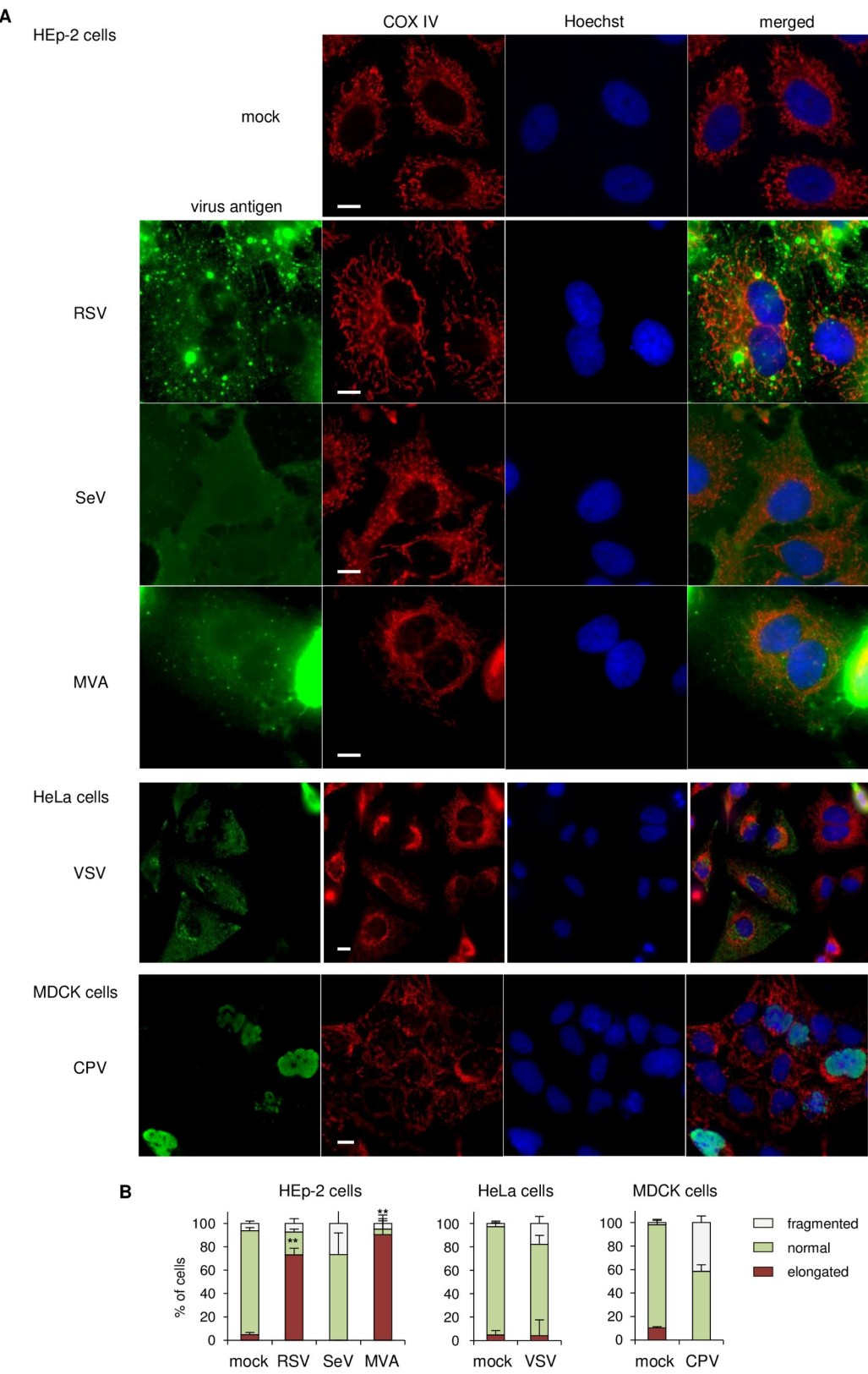

**Fig 6. RNA and DNA viruses that cause downregulation of mitochondrial biogenesis induce mitochondrial hyperfusion.** (A) Confocal microscopy images of cells infected with RSV, SeV, MVA (HEp-2 cells), VSV (HeLa cells), and

CPV (MDCK cells). Mitochondria were stained with anti COX IV antibody (red) and virus antigen was stained with antibody against each virus protein (green), as described in Materials and Methods. Scale bar = 10 μm. (B) Quantification of the impact of virus infection on mitochondria morphology. The mitochondrial network of ~30 cells per condition and experiment (*n* = 3) were classified into normal, fragmented, and elongated morphological categories. Data are the mean value ± SD (*n* = 3). Statistical significance was determined using an unpaired Student's *t*-test; $^{**}P < 0.01$.

by Mfn1 (Fig 3A–3C), which resulted in the release of mtDNA, as seen with HSV-1. Furthermore, we showed that other cellular stress conditions such as UV or ActD cause mitochondrial hyperfusion but do not cause the release of mtDNA (Fig 3E–3G). Therefore, it is suggested that mtDNA release followed by mitochondrial hyperfusion induced by virus infection is required for undetermined inherent processes, and further studies are required to clarify the phenomenon.

Recent reports showed that full protection against diverse RNA viruses also relies on STING. Lack of STING significantly reduced the production of type I IFN and resulted in failure to mount a strong innate immune response against RNA viruses such as VSV and SeV [53]. Single-stranded positive-sense RNA (+ssRNA) virus families, flaviviruses and coronaviruses, have developed mechanisms to block STING-dependent signaling, in which the viral proteins can function as antagonists of the signaling [54]. From these reports, it has become increasingly apparent that STING also plays an important role in restricting RNA virus replication. However, these studies confirmed that cGAS was not implicated in the reactions against -ssRNA viruses [26,53].

There are some reports that certain +ssRNA viruses are affected by cGAS in host innate immunity. For example, cGAS inhibits the replication of various +ssRNA viruses such as flaviviruses and alphaviruses [55,56]. cGAS knockout mice were more susceptible to infection with West Nile virus, a member of the flaviviruses [55]. It is speculated that cGAS plays a role in maintaining the basal level of ISG expression, which suppresses virus replication. Intriguingly, recent reports revealed that dengue virus and Zika virus, which belong to the family Flaviviridae, induce mtDNA release into the cytosol, which is captured by cGAS [57,58]. These results indicate that host cells employ cGAS as a defense strategy against +ssRNA viruses, although the trigger of mtDNA release has not yet been identified.

In the present study, we propose that induction of antiviral priming by mtDNA via cGAS is a general response to virus infection, which can lead to the intrinsic downregulation of mitochondrial protein expression. However, among the viruses we tested, MeV, RSV and MVA, no common viral component or characteristic was found, thus the factor(s) responsible for mitochondrial downregulation is still unclear. In the present study, we utilized PGC-1α knockdown as a mimic of the downregulation of mitochondrial biogenesis, and showed that PGC-1α depletion caused mtDNA release and IFN-β induction (Fig 3I and 3J). In addition, as described above, depletion of TFAM also induces mtDNA liberation and a consequent antiviral response due to mtDNA stress [24]. However, the PGC-1α and TFAM expression levels were not altered by MeV infection, as determined by RT-qPCR (S4C Fig), and were not included in the list of downregulated genes post-infection for all of these viruses. Thus, mitochondrial downregulation by these viruses was not caused by a decrease in the amount of PGC-1α or TFAM directly, but targeted the host transcriptional network for mitochondrial biogenesis. Interestingly, all six viruses in the present study caused the downregulation of expression of over 100 genes; however, SeV, VSV and CPV showed little or no enrichment in gene ontology analysis, indicating that the strategy of downregulation was no operating in these viruses. By contrast, the other viruses targeted various cellular components in addition to mitochondria, thus they might possess an individual approach for downregulation. It is difficult to clarify comprehensively the inherent mechanism by which the virus targets specific

cellular pathways and induces alterations in the host transcriptional regulatory network. We are currently attempting to depict the whole transcriptional regulatory network of the host after MeV infection using CAGE (cap analysis of gene expression) with a next generation sequencer and comprehensive analysis of promoter activities across the whole genome. These high-throughput experiments may uncover the comprehensive host response to virus infection, and will be useful for understanding the whole picture regarding the complexity of virus infection in general.

## Materials and methods

### Ethics statement

All animal experiments were approved by the Animal Experiment Committee at The University of Tokyo (approval number; PA17-30) and were performed in accordance with the Regulations for Animal Care and Use of The University of Tokyo.

### Antibodies, cells and viruses

Antibodies used in this study are described in S1 Table. 293SLAM cells (HEK293 cells stably expressing an MeV receptor SLAM) [13], Vero cells (from ATCC), Vero-hSLAM cells (Vero cells stably expressing SLAM) and MCF7 cells (human breast cancer cells; from the Cell Resource Center for the Biomedical Research Institute of Development, Aging and Cancer, Tohoku University, Miyagi, Japan) were maintained in Dulbecco's modified essential medium (DMEM) supplemented with 5% fetal calf serum (FCS). HEp-2 cells (human laryngeal carcinoma cells; from ATCC) and MDCK cells (Madin–Darby canine kidney cells; from ATCC) were cultured in DMEM supplemented with 10% FCS. NCI-H441 cells (human non-small-cell lung cancer cells; from JCRB cell bank, National Institute of Biomedical Innovation, Japan) or B95a cells (marmoset lymphoblastoid cells; kindly gifted from Dr Kobune) [59] were maintained in RPMI 1640 medium supplemented with 10% or 5% FCS, respectively. A72-B cells (canine fibroblast cells; kindly provided from Intervet KK Co. Ltd., Japan) were cultured in Eagle's minimal essential medium supplemented with 10% FCS and 0.3% triptose phosphate broth. MeV-HL and rMV-EGFP [18] or MeV-CAMR40 [41] were infected with B95a cells or Vero cells supplemented with 2% FCS, respectively, and harvested by freeze-thawing and sonication when syncytium formation was at its maximum. Strain RSV-A2 was inoculated into HEp-2 cells supplemented with 2% FCS, and 4 days later cells were harvested together with culture medium and sonicated. After centrifugation at $1,750 \times g$ for 10 min, the supernatant was collected and supplemented with 3.5% (v/v) dimethyl sulfoxide. SeV strain Z was inoculated into 10-day-old embryonated chicken eggs. After 3 days of incubation, the allantoic fluid was harvested and assayed for plaque forming units (pfu) as described previously [60]. MVA (MVA-T7; which is encoded by the T7 polymerase gene) was grown in primary chicken fibroblasts [61]. Strain CPV-JJ-C-13 was inoculated into A72-B cells, and the culture medium at 7 days post-infection (dpi) was centrifuged at $1,750 \times g$ for 10 min, and the supernatant was collected. The virus titer was measured with MDCK cells. Viruses were infected into cells at a multiplicity of infection of 0.01 (MeV-HL, rMV-EGFP, VSV), 0.1 (RSV, SeV, MVA) or 0.25 (CPV).

### Fluorescence microscopic observations

MitoTracker Orange (Invitrogen, USA) and Hoechst33342 (Cambrex Bio Science Walkersville, USA) were added to the cell culture medium at 0.5 μM and 2 μg/ml, respectively, and incubated for 10 min at 37˚C. Cells were washed twice with fresh culture medium and further

incubated for 30 min. Alternatively, cells were fixed with 3% paraformaldehyde in PBS for 30 min, permeabilized with 0.5% Triton X-100 in PBS for 20 min, then reacted with primary antibodies for 60 min, and stained with secondary antibodies for 60 min at room temperature. Cells were washed with PBS between each step. Fluorescence was observed with a BZ-X700 laser confocal microscope (KEYENCE, Japan). Image processing (e.g. black balance) was performed using a BZ-X Analyzer BZ-H3A (KEYENCE). Other image processing, such as haze reduction and edge enhancement, was not applied. The mitochondrial network was examined and classified into three distinct categories; normal, fragmented, and elongated morphology. A mitochondrial network was considered elongated when the elongation was observed on the periphery of the nuclei with at least five mitochondria longer than 15 μm (corresponding to the approximate diameter of nucleus) and/or displayed interconnected branched morphology. Mitochondria were classified as fragmented when $\geq$ 75% of mitochondria appeared in a dot-like pattern with no signs of elongation. For time-lapse imaging of mitochondrial morphology, Vero-hSLAM cells seeded in 35 mm glass bottom dishes were infected with rMV-EGFP and cultured in a Stage Top Incubator system (INU-ONICS and GM2000; TOKAI HIT, Japan) at 37˚C in 5% $CO_2$. After 10 h, cells were stained with MitoTracker as described above. From 11 h post-infection, fluorescence images were acquired every 6 min for 20 h using an IX70 laser confocal microscope and the FluoView FV1000 system (Olympus, Japan) at 550 V for green and 600 V for red.

## RNA interference

Specific siRNAs (S1 Table) were purchased from Thermo Fisher Scientific (USA). Cells in 24-well plates were transfected with 6 pmol of the siRNA using Lipofectamine RNAiMAX reagent (Invitrogen, USA), according to the manufacturer's instructions.

## Quantitative PCR

Total RNA was extracted from the cells lysed with ISOGEN (NipponGene, Japan), and was reverse-transcribed with PrimeScript Reverse Transcriptase (Takara, Japan) and oligo(dT) primer and random primer (6-mer) according to the manufacturer's instruction. qPCR was performed with THUNDERBIRD SYBR qPCR mix (Toyobo, Japan) and specific primers described in S1 Table. qPCR assays were conducted on a Rotor-Gene Q cycler (Qiagen, German) or LightCycler 96 (Roche, Switzerland). Three technical replicates were performed for each biological sample, and expression values of each replicate were normalized against cDNA of 28S rRNA using the 2-ΔΔCT method. For relative expression (fold), control samples were centered at 1.

## Production of mammalian expression plasmids

To construct HA-tagged human cGAS and Mfn1, cDNAs of their open reading frames were amplified with specific primer pairs flanking *Sal*I/*Not*I restriction sites (5′-gtcgaccatgcagccttgg-cacggaaag-3′ and 5′-gcggccgctcaaaattcatcaaaaactggaaactcattg-3′) or *Eco*RI/*Not*I restriction site (5′-gaattcggatggcagaacctgtttctcc-3′ and 5′-gcggccgcttaggattcttcattgcttgaagg-3′), respectively, using a reverse transcription product of total RNA in MCF7 cells and LA-Taq DNA polymerase (Takara). Obtained PCR products were introduced into pCMV-HA (Clontech, USA). To construct EGFP-fused human LC3B, LC3B cDNA was amplified with a primer pair flanking a *Bgl*II restriction site (5′-agatctatgccgtcgcggagaagaccttc-3′ and 5′-agatctggtaccgcggccgccttacactga-caatttcatcccg-3′) in the same manner as above, and then inserted into pEGFP-C1 (Clontech). To construct expression plasmids encoding the MeV-F and H genes, cDNAs were amplified

with the appropriate primer pairs, and then inserted into the pCAGGS mammalian expression vector [62].

## Depletion of mtDNA and detection of mtDNA in cytosolic extracts

Cells were treated with 100 μM of ddC for 4 days. To observe mtDNA in cells, 1/200 volume of PicoGreen dsDNA reagent (Molecular Probes Life Technologies, USA) was added to the cell culture medium and incubated for 60 min. After washing twice with fresh medium, fluorescence was observed as described above. To measure the efficiency of mtDNA depletion, total extracts were prepared by resuspending the cells in 50 mM NaOH, followed by incubation at 95°C for 30 min, and neutralization by adding a 10% volume of Tris-HCl (pH 7.5). To measure cytosolic mtDNA, cells were divided into two equal aliquots, and one aliquot served as the total mtDNA described above or the whole cell lysate (WCL) for western blotting. The second aliquot was resuspended in 200 μl of buffer containing 150 mM NaCl, 50 mM HEPES pH 7.4, and 25 μg/ml digitonin, incubated with inversion for 10 min to allow selective plasma membrane permeabilization, and was then centrifuged at $1,000 \times g$ for 3 min. The pellet was used as the "ppt" fraction for western blotting. The cytosolic supernatants were transferred to fresh tubes and spun at $17,000 \times g$ for 10 min to pellet any remaining cellular debris. Prepared samples were subjected to qPCR of a specific region in the D-loop of mtDNA as described below, or were alternatively used as the "cyto" fraction for western blotting.

## Immunoprecipitation/PCR

MCF7 cells transfected with HA-cGAS plasmid were mock-treated or infected with rMV-EGFP. After 24 h, cells were crosslinked with 10% formaldehyde in PBS at 37°C for 10 min, washed twice with PBS, lysed with cell lysis buffer (0.5% TritonX-100/0.5 mM EDTA/PBS supplemented with protease inhibitor cocktail (Thermo Scientific) and incubated at 4°C for 10 min, and centrifuged at $18,000 \times g$ for 10 min. The supernatant was divided into two equal aliquots, and 1 μg of control antibody (mouse IgG1 isotype) or mouse anti-HA tag antibody was added, respectively, along with 20 μl of protein-G sepharose slurry (GE Healthcare, USA), and incubated with inversion for 4 h. The resin was washed four times with cell lysis buffer, four times with TE (10 mM Tris-HCl, pH 7.5, 1 mM EDTA), and incubated with 0.5% SDS/130 μg/ml proteinase K (Roche) in TE at 55°C for 6 h. The supernatant was extracted with phenol:chloroform and then with chloroform, and the aqueous phase was collected, precipitated with ethanol supplemented with 50 mg/ml glycogen. The obtained pellet was dissolved in TE and subjected to qPCR for mtDNA, as described above. The value of the control antibody was subtracted from that of the anti-HA-tag antibody, and the obtained values were used for enrichment analysis.

## Flow cytometry

A total of $1 \times 10^6$ cells were washed with PBS and trypsinized in 0.025% trypsin/0.24 mM EDTA. After pelleting by centrifugation at $2,000 \times g$ for 5 min, the cells were fixed with Fixation Buffer (BioLegend, USA) at room temperature for 20 min in the dark. The cells were then washed with PBS containing 2% FCS, and then stained with 100 nM of MitoGreen (PromoKine, Germany). Cells were washed twice with PBS containing 2% FCS, and the intensity of fluorescence was measured using an Attune NxT Acoustic Focusing Cytometer (Invitrogen). To determine the extent of the mitochondrial mass, the mean fluorescent intensities of the cells stained with MitoGreen were calculated using Flowjo software ver.10 (Nihon Becton Dickinson, Japan), and the mock sample was centered at 1. The staining relies on the mitochondria mass, not on mitochondrial membrane potential.

## Measurement of mitochondrial membrane potential

MCF cells were cultured in DMEM without phenol red in 96-well plates, and were infected with MeV-HL for 1, 2 and 3 days, or were alternatively treated with 1 μM of CCCP (FujiFilm, Japan) or 10 μM of rotenone (Sigma, USA), for 4 h. The levels of mitochondrial membrane potential were measured using a JC-10 mitocondiral membrane potential assay kit (Abcam, UK) according to the manufacturer's instruction. The JC-10 dye exhibits two staining spectra. In normal respiring cells, the dye forms aggregates in the mitochondrial membrane, exhibiting orange fluorescence. However, when membrane potential is lost, monomeric JC-10 forms in the cytosol, exhibiting green fluorescence. The fluorescent intensities of cells at excitation/emission wavelengths of 540/590nm (orange) and 490/525nm (green) were measured using the GloMax discover/explorer system (Promega). Fluorescence intensities were described as the 590/520 ratio. For the relative membrane potential (fold-value), the control sample was centered at 1.

## Luciferase reporter assay

MCF cells in 24-well plates were transfected with 0.9 μg of pISRE-Luc plasmid, which expresses firefly luciferase (Fluc) (Clontech) and 0.1 μg of phRL-TK (Int-) plasmid which expresses renilla luciferase (Rluc) (Promega) using Lipofectamine LTX Transfection Reagent (Life Technologies, USA). The phRL-TK (Int-) plasmid, driven by a thymidine kinase promoter, was used as an internal control for transfection efficiency. At 24 h post-transfection, the medium was replaced with fresh medium, and then MeV-HL was added, and incubated at 37˚C. At the indicated time, as shown in Fig 2D, cells were harvested and the luciferase activities were measured using a Dual-Luciferase Reporter Assay System (Promega) according to the manufacturer's instructions. The Fluc/Rluc value was calculated and the 0 h sample was centered at 1.

## WST-1 assay

Cell viability was determined using a WST-1 Cell Proliferation kit (Takara) at 1, 3, 5, and 7 dpi, according to the manufacturer's protocol. For relative cell viability, mock-treated cells were centered at 1.

## Mice

$MAVS^{-/-}$ mice (B6.129B6-Mavs<tm1Tsse>) with a C57BL/6 background were initially provided as frozen sperm by RIKEN BRC through the National Bio-Resource Project of the MEXT, Japan. $cGAS^{-/-}$ mice (B6(C)-$Mb21d1^{tm1d(EUCOMM)Hmgu}$/J) with a C57BL/6 background were purchased from The Jackson Laboratory (USA). C57BL/6 mice (wild-type) were purchased from Nihon CLEA (Japan). Mice were kept and bred under specific-pathogen-free conditions in negative pressure isolators.

## Animal infection experiments

Three-week-old mice with body weight > 7 g were inoculated with $1 \times 10^4$ TCID$_{50}$ in 30 μl of MeV-CAMR40 into the left cerebral hemisphere. A survival curve was measured until 13 dpi, alternatively mouse brains were collected at 5 dpi and homogenized with ISOGEN for reverse transcription (RT)-qPCR as described above.

### Microarray data

Microarray data of MeV were deposited in GEA, accession number E-GEAD-331 (https://ddbj.nig.ac.jp/public/ddbj_database/gea/experiment/E-GEAD-000/E-GEAD-331/) [15]. The GEO (https://www.ncbi.nlm.nih.gov/geo/) accession numbers for microarray data utilized in this study are GSE32140 for RSV [42], GSE38866 for VSV [43], GSE53103 for SeV [44] and GSE72397 for CPV [47]. Microarray data of MVA referred to a previous report [44].

### Gene ontology annotation

GO annotation of the downregulated genes post-infection was performed at the Gene Ontology Consortium (http://geneontology.org/) with analysis type: PANTHER Overrepresentation Test (Released 20171205), Annotation version: GO ontology database (released 2018-08-09), Annotation data set: GO cellular component complete, Test type: Binomial (used the Bonferroni correlation for multiple testing) [63–65].

### Statistical analyses

Data shown are representative of three independent experiments. Error bars displayed throughout the manuscript represent standard deviations, and were calculated from duplicate or triplicate technical replicates of each biological sample. Statistical significance was determined using unpaired Student's $t$-tests or one-way analysis of variance (ANOVA) followed by Dunnett's multiple comparison test; $^*P < 0.05$; $^{**}P < 0.01$; ns, not significant ($P > 0.05$). For mouse survival curves, statistical significance was determined by the log-rank test.

## Supporting information

**S1 Fig. MeV infection induces downregulation of mitochondrial biogenesis.** (A) List of gene ontology analysis results of the cellular components overrepresented among genes downregulated by MeV infection. Uppermost layers from the analysis are described. The entire list, including all layers and their values, are described in S2 Table. (B) MCF7 cells were infected with MeV for 3 days. The mitochondrial mass in the cells was analyzed with MitoGreen staining and flow cytometry. The mean fluorescence intensity of MitoGreen was measured as described in Materials and Methods. (C) MCF7 cells were infected with MeV for 1, 2 and 3 days, or treated with 10 μM of CCCP or 1μM rotenone for 4 h. The mitochondrial membrane potential was measured with JC-10 dye as described in the Materials and Methods, and it was described as the fluorescence intensities at an $A_{590}/A_{520}$ ratio. (D) Total amount of mtDNA in cells was quantitated by qPCR as described in the Materials and Methods. (E) Vero-hSLAM cells were mock-transfected or co-transfected with MeV-F and H expression plasmids. After 1 day, cells were fixed and the mitochondria and nuclei were stained with anti-COX IV antibody and Hoechst, respectively, as described in the Materials and Methods. Multinuclear giant cells induced by expression of MeV-F and H proteins are indicated by a white dotted line. Scale bar = 10 μm. The mitochondrial morphology of mock-transfected cells or multinuclear giant cells (~20 cells per conditions) in three experiments was classified as a normal, elongated, or fragmented mitochondrial network (right panel). Data are the mean value ± SD ($n = 3$). Statistical significance was determined using an unpaired Student's $t$-test; $^*P < 0.05$; $^{**}P < 0.01$; ns, not significant ($P > 0.05$).
(EPS)

**S2 Fig. MeV infection induces mitochondrial hyperfusion.** (A) Confocal microscopy images of H441 cells and MCF7 cells infected with rMV-EGFP. Mitochondria and nuclei were stained with MitoTracker and Hoechst, respectively, at 24 hpi. Mitochondria morphology of at least

40 cells per condition and in three independent experiments were classified into three groups; normal, elongated, and fragmented mitochondrial network (right panel). Scale bar = 10 μm. Data are the mean value ± SD ($n = 3$). Statistical significance was determined using an unpaired Student's $t$-test; $**P < 0.01$. (B) Vero-hSLAM cells were transfected with plasmid expressing LC3-EGFP. One day later, cells were infected with strain MeV-HL and fixed with paraformaldehyde at 16 hpi. Mitochondria or MeV-N protein were stained with anti-COX IV monoclonal antibody or anti-MeV-N rabbit polyclonal antibody, respectively, as described in Materials and Methods. Scale bar = 10 μm. Lower images are enlargements of squared region. (EPS)

**S3 Fig. The role of cGAS in innate antiviral responses against MeV.** (A) MCF7 cells mock-treated (left panel) or infected with MeV (right panel) were subjected to digitonin fractionation as described in the Materials and Methods and whole cell lysate (WCL), pellets (ppt) or cytosolic extracts (cyto) were blotted using the indicated antibodies. (B) MCF7 cells transfected with siRNA for the NC or cGAS were infected with MeV, and the RNA collected at 24 hpi was subjected to RT-qPCR for quantification of seven ISGs (left panel) or three housekeeping genes (right panel). Data are representative of three independent experiments. (C) Vero-hSLAM cells transfected with siRNA for the NC or MAVS were infected with rMV-EGFP. At 16 hpi, cells were fixed and the mitochondria and nuclei were stained with anti-COX IV antibody and Hoechst, respectively, as described in Materials and Methods. Scale bar = 10 μm. The mitochondrial morphology of cells transfected with siRNA for MAVS (~40 cells per conditions) in three experiments was classified as a normal, elongated, or fragmented mitochondrial network (right panel). (D) MCF cells were treated with the mock control or ddC for 3d, and then subjected to WST-1 assay for measurement of cell viability (left), or infected with MeV and the virus titer at 2 dpi was determined (right). Data are the mean value ± SD ($n = 3$). Statistical significance was determined using an unpaired Student's $t$-test; $*P < 0.05$; $**P < 0.01$; ns, not significant ($P > 0.05$). (EPS)

**S4 Fig. Effect of knockdown of Mfn1 or PGC-1α.** (A) MCF7 cells transfected with siRNA for Mfn1 were subjected to digitonin fractionation as described in the Materials and Methods and whole cell lysate (WCL), pellets (ppt) or cytosolic extracts (cyto) were blotted using the indicated antibodies. (B) Vero-hSLAM cells transfected with siRNA for the NC or Mfn1 were treated with 3 μg/ml ActD or 60 mJ/cm$^2$ UV-C. Mitochondria and nuclei were stained 7 h later with MitoTracker and Hoechst, respectively. Scale bar = 10 μm. The mitochondrial morphology of cells transfected with siRNA for Mfn1 (~20 cells per conditions) in three experiments were classified as normal, elongated, or fragmented mitochondrial network (right panel). (C) MCF7 cells were infected with MeV, and the total RNA at 3 dpi was subjected to RT-qPCR for quantification of PGC-1α and TFAM. (D) Total RNA extracted from mock (NC) or PGC-1α knockdown cells was subjected to RT-qPCR for quantification of five nuclear genes encoding mitochondrial proteins (upper left panel), three non-mitochondrial protein genes (upper right panel), or seven ISGs (lower panel). (E) MCF7 cells were transfected with siRNA for the NC or PGC-1α, and the mitochondrial membrane potentials were measured as described in S1C Fig. (F) MCF7 cells transfected with siRNA for PGC-1α were subjected to digitonin fractionation same as (A). Data are representative of three independent experiments. Data are the mean value ± SD ($n = 3$). Statistical significance was determined using an unpaired Student's $t$-test; $*P < 0.05$; $**P < 0.01$; ns, not significant ($P > 0.05$). (EPS)

**S5 Fig. Requirement for cGAS in controlling viral infection in vivo.** Wild type or *cGAS*$^{-/-}$ mice ($n = 8$, male = 3, female = 5) were intracerebrally inoculated with $1.0 \times 10^3$ TCID$_{50}$ of MeV-CAMR, and the cerebrum was harvested at 5 dpi. The amounts of mRNA of eight ISGs measured by RT-qPCR are represented relative to the amount of MeV-N. Statistical significance was determined using an unpaired Student's *t*-test.
(EPS)

**S6 Fig. Gene ontology analysis of the cellular components overrepresented among genes downregulated by RSV, SeV, VSV, and MVA infection.** Uppermost layers from the analysis are described. The entire list including all layers and values is presented in S2 Table.
(EPS)

**S1 Table. List of antibodies, siRNAs and primer pairs used in this study.**
(PDF)

**S2 Table. Entire lists of the results of gene ontology analysis.**
(PDF)

**S1 Movie. Dynamics of syncytia formation in MeV-infected cells.** Vero-hSLAM cells seeded in glass-bottom dishes were infected with rMV-EGFP and stained with MitoTracker at 10 hpi. Live cell imaging of EGFP started at 8 hpi and proceeded every 6 min for 13 h.
(MP4)

**S2 Movie. Dynamics of mitochondrial morphology in MeV-infected cells.** Vero-hSLAM cells seeded in glass-bottom dishes were infected with rMV-EGFP and stained with MitoTracker at 10 hpi. Live cell imaging of MitoTracker started at 8 hpi and proceeded every 6 min for 13 h. Characteristic entangled foci of mitochondria are indicated by circles.
(MP4)

**S3 Movie. Dynamics of mitochondrial morphology in syncytia in the MeV-infected cells.** Vero-hSLAM cells seeded in glass-bottom dishes were infected with rMV-EGFP and stained with MitoTracker at 10 hpi. Live cell imaging of MitoTracker merged with EGFP started at 8 hpi and proceeded every 6 min for 13 h.
(MP4)

## Author Contributions

**Funding acquisition:** Chieko Kai.

**Investigation:** Hiroki Sato, Miho Hoshi, Fusako Ikeda, Tomoko Fujiyuki.

**Methodology:** Misako Yoneda.

**Project administration:** Chieko Kai.

**Writing – original draft:** Hiroki Sato.

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
