## [Decision Letter · Decision Letter 0]

15 Jun 2020

Dear Prof. Kai,

Thank you very much for submitting your manuscript "Downregulation of mitochondrial biogenesis by virus infection triggers antiviral responses by cyclic GMP-AMP synthase" for consideration at PLOS Pathogens. As with all papers reviewed by the journal, your manuscript was reviewed by members of the editorial board and by several independent reviewers.

Both reviewers thought the observation you made is fascinating and worthwhile to investigate. However, they thought there is some lack of evidence whether this is only a side-effect of MeV-induced CPE or direct (and new) antiviral mechanism. I would encourage you to follow the guidelines of the reviewers to improve the study, and we will be happy to reconsider it when you decide to include more evidence in support of your hypothesis.In light of the reviews (below this email), we would like to invite the resubmission of a significantly-revised version that takes into account the reviewers' comments.

We cannot make any decision about publication until we have seen the revised manuscript and your response to the reviewers' comments. Your revised manuscript is also likely to be sent to reviewers for further evaluation.

Sincerely,

Matthias Johannes Schnell, PhD

Associate Editor

PLOS Pathogens

Benhur Lee

Section Editor

PLOS Pathogens

Kasturi Haldar

Editor-in-Chief

PLOS Pathogens

orcid.org/0000-0001-5065-158X

Michael Malim

Editor-in-Chief

PLOS Pathogens

orcid.org/0000-0002-7699-2064

Reviewer's Responses to Questions

**Part I - Summary**

Reviewer #1: In their study entitled “Downregulation of mitochondrial biogenesis by virus infection triggers antiviral responses by cyclic GMP-AMP synthase” Sato et al. provide data that infection with measles virus (MeV), an RNA virus, causes mitochondrial elongation associated with release of mtDNA into the cytosol. Furthermore, they show that in addition to typical RNA innate immune activation, that the mtDNA release is accompanied by activation of the cytosolic DNA sensor cGAS, both in vitro and in vivo. Host cells infected with MeV show a downregulation of mitochondrial respiration and matrix genes, which the authors interpret as reduced mitochondrial biogenesis, and then show that knock-down of PGC-1a is sufficient to cause mtDNA release and cGAS-IFN signaling in MCF7 cells. They then determine that some additional, but not all, RNA and DNA viruses lead to downregulation of mitochondrial genes and induction of mtDNA-mediated activation of cGAS-STING pathway. This is a very interesting and important study. If true, this is the first implication of mitochondrial biogenesis per se in mtDNA-mediated innate immune signaling and implicating mtDNA as an agonist of host anti-viral signaling for RNA viruses more broadly is important. However, additional data and experiments are needed to solidify the authors’ main conclusions.

Reviewer #2: Sato et al. describe in this manuscript the involvement of the cytoplasmic DNA sensor cGAS in interferon induction and setup of an antiviral state in response to an infection with an RNA virus, measles virus (MeV). Upon infection they observe morphological changes in mitochondria leading to the release of mitochondrial DNA into the cytoplasm. These DNA molecules then activate cGAS and lead to increased IFN production. The authors provide mechanistic insights into the molecular processes by employing siRNA knock-down and overexpression strategies for cGAS and genes involved in mitochondria function, Mfn1 and PGC-1alpha. They then show that cGAS contributes to establishment of an antiviral response against MeV in an animal model. Finally, the authors extend their observations to several other RNA and DNA viruses. Importantly, while the pneumovirus respiratory syncytial virus (RSV) showed cGAS-dependent IFN induction, the murine paramyxovirus Sendai virus (SeV) and the rhabdovirus vesicular stomatitis virus (VSV) showed no effects of cGAS. Similarly, modified vaccinia virus Ankara (MVA), a dsDNA poxvirus, activated cGAS through this mitochondrial pathway, whereas a ssDNA virus, canine parvovirus (CPV) did not.

The contribution of cGAS to antiviral responses against negative-strand RNA viruses (NSV) is novel. The authors suggest that this contribution is a specific cellular response to certain NSV. Although a common feature of viruses involving this pathway is downregulation of gene expression of genes involved in mitochondria biogenesis, the panel of viruses tested leaves a big question mark on why some viruses exhibit this phenotype (MeV, RSV, MVA), while others do not (VSV, SeV, CPV).

From the immunofluorescence stainings provided here it becomes apparent that MeV infection induces mitochondrial hyperfusion leading to release of mitochondrial DNA into the cytoplasm. The authors suggest that this event is a specific antiviral response. However, it may as well be a manifestation of virus-induced cell death. The authors use a wild type MeV which has a highly fusogenic membrane fusion apparatus consisting of H and F proteins. The virus induces the formation of very large syncytia, which are known to change morphology and rapidly lose integrity. Mitochondrial hyperfusion and release of mtDNA may be an indirect consequence of the overall loss of cell integrity. The use of Actinomycin D and UV irradiation, which induce mitochondrial stress but not cytosolic mtDNA release, support this hypothesis.

While the authors use specific siRNA knock-down approaches to show synergy between the RIG-I-like receptor/MAVS-mediated and the cGAS-mediated IFN induction in response to MeV infection, an apparent question arises whether MAVS-mediated signaling processes are involved in changing mitochondria morphology. Since MAVS is incorporated into the outer mitochondrial membrane, assembly of the RLR signaling complex at these sites may play a role in mitochondria hyperfusion and mtDNA release. Unfortunately, while possibly important in the context of this study, this is addressed neither in experiments nor in the discussion of this manuscript.

The mouse experiments finally show some contribution of cGAS to MeV pathogenesis. The authors choose a model of intracranial infection of MeV into wild type, MAVS-/- and cGAS-/- mice. Surprisingly, while the authors claim to have a lethal infection model (line 204), 60% of wild type mice survive this infection. Notably, in the cited publication, MeV infection using the same virus and mouse strains caused 100% lethality (Arai et al. J Gen Virol. 2017;98(7):1620-9). Or was it intentional to use sublethal infections in wild type mice?

**Part II – Major Issues: Key Experiments Required for Acceptance**

Reviewer #1: 1) The authors main conclusion, which is even reflected in the title, is that downregulation of “mitochondrial biogenesis” is the key feature that drives mtDNA release and subsequent cGAS activation during MeV infection. However, the authors never actually never measure or perturb mitochondrial biogenesis in the infected cells. They instead observe that the mitochondria are hyperfused and show that reducing fusion by depletion of MFN1 reduced mtDNA-mediated cGAS signaling. This experiment shows that “mitochondrial dynamics” is involved, not “mitochondrial biogenesis.” Thus, the conclusions about biogenesis are based mainly, if not exclusively, on the gene signatures observed. The authors need to better characterize the mitochondrial phenotype in the infected cells, with a goal of demonstrating an effect on biogenesis. For example, is mitochondrial abundance lower in the infected cells, as predicted if biogenesis is inhibited. This can be measured with comparative westerns of mitochondrial and other housekeeping genes, by flow cytometry with mitochondrial potential-independent dyes, etc. Given the gene signature includes respiratory genes, the authors should also address mitochondrial membrane potential and OXPHOS activity, which should also be downregulated if biogenesis is reduced. Finally, is PGC-1a, TFAM and other or other biogenesis genes downregulated in the infected cells at the RNA and protein level? Finally, can overexpression of PGC-1a in the infected cells prevent mtDNA-mediated signaling as predicted by their model?

2) Related to point #1 above, to implicate mitochondrial biogenesis as sufficient for mtDNA-mediated signaling, the authors do knock-down PGC-1a and show that this instigates mtDNA release and cGAS signaling. To my knowledge this is the first time that reducing this pathway has been shown to be sufficient to mediate mtDNA innate immune signaling. Accordingly, this needs to be iron clad. First, similar direct measures of mitochondrial biogenesis and activity (not just qPCR of genes) needs to be demonstrated in this experiment (as described above) to show that the KD of PGC-1a leads to lower mitochondrial mass as predicted. Also, does this lead to mitochondrial hyperfusion as seen in the MeV-infected cells, which would corroborate in the KD and infection models? In addition, why was this experiment done in MCF7 cells? To make a stronger conclusion that reduced mitochondrial biogenesis is sufficient for mtDNA release and innate immune sensing, other cell types, including non-transformed cells or immune cells, should be tested. Finally, does KD of PGC-1a alone provide an anti-viral effect as predicted?

3) The authors use cellular fractionation and cGAS immunoprecipitation to demonstrate the presence of mtDNA in the cytosol. First, it is critical that the authors show the western blots demonstrating the purity of the fractions used for this experiment in all cases (i.e. during infection and PGC-1a KD, etc.). Second, others have shown that mtDNA release involves enlarged or clustered mtDNA nucleoids and mtDNA depletion. The authors should visualize mtDNA with picogreen or an anti-DNA antibody to address nucleoid morphology and perhaps also see it outside of mitochondria as predicted, and assess mtDNA copy number, which should also be depleted if mitochondrial biogenesis is downregulated.

4) The authors propose “a two-step induction of antiviral responses; at an early phase of infection, viral RNA replication is detected rapidly by an RNA sensor, while during the late phase of infection, mitochondrial downregulation accompanied by mtDNA liberation causes prolonged IFN-� and ISG production. These findings uncover a novel host strategy of the defense system for suppressing viral propagation.” This is a really important concept, thus, can the authors perform a time course of MeV infection, in which they can show that activation of the RNA sensor pathways precedes that of the mtDNA sensing pathway?

Reviewer #2: 1. Influence of MAVS pathway on mitochondrial hyperfusion and cGAS activation: Is mitochondrial hyperfusion after MeV infection dependent on RLR-signaling? This could easily be addressed by measuring mitochondrial hyperfusion in MAVS knock-down cells vs. wild type cells.

2. Dependence on syncytia formation: The authors should provide a more detailed kinetic of mitochondria hyperfusion and correlate their findings with the kinetics of syncytia formation. Also, comparison of wild type and vaccine MeV strains, which have different fusogenic potentials, will offer valuable insights. Alternatively, treatment of cells with fusion imhibitory peptide after MeV infection would address this question.

3. Mouse experiments: Although introduced as a lethal infection model (line 204), 60% of wild type mice survive infection by day 13. Why was the experiment terminated at this time? Would the data and their interpretation have changed if mice were observed for a longer period, i.e. would more mice have succumbed to infection after day 13? Also, according to materials and methods, the authors injected a total volume of 30 ul intracranially into mice with at least 7 grams body weight. This seems a very high amount. Could this volume be considered to induce traumatic brain injury by itself? How was it injected: Free-hand or using a stereotactic device and microfluid pump to control injection flow rate?

**Part III – Minor Issues: Editorial and Data Presentation Modifications**

Reviewer #1: • In the discussion the authors use a term called “mitochondrial thickness”. This term is undefined and not very clear from the images presented (e.g. Fig 1B).

• In Figure 3D (cytosolic mtDNA measurement) is the difference between mock and MeV under control or Mfn 1 downregulation significant? Please indicate statistical analysis used.

• Figure 4A: X and Y axes are no labelled.

• In Figure 5B, SeV and MVA infection seems to be dramatically lower than that of RSV, probably explaining the lack of mitochondrial elongation? Is this confounding?

• Scale bars should be indicated in Figures 2D, 3E, 5B, S1B & C.

Reviewer #2: Fig. 2B: What are the absolute levels of mtDNA enrichment in contrast to the presented relative enrichment values?

Fig. 2C: After MeV infection, are the IFNbeta mRNA levels in MAVS-cGAS double-knockdown cells significantly lower than MAVS knockdown alone? The indicated asterisks seem to compare each sample to NC control.

Figs. 2 and 5: Use of ddC: What are the effects of blocking mtDNA replication on cell survival and virus replication? If mitochondria are not functioning properly this may affect cell viability and virus replication in general.

Fig. 5B: No images of uninfected HeLa and MDCK cells are provided to assess mitochondria structures.

S3 Figure: The labels of the subpanels do not align with what is referred to in the text (Is S3B supposed to be the diagram next to the IF images? Is S3C the bar diagram on the bottom or all three bar diagrams labelled as S3B?) Please correct.

lines 171-175: This sentence is confusion. Please edit.

PLOS authors have the option to publish the peer review history of their article (what does this mean?). If published, this will include your full peer review and any attached files.

Reviewer #1: No

Reviewer #2: No
---

## [Decision Letter · Decision Letter 1]

28 Jun 2021

Dear Prof. Kai,

Thank you very much for submitting your manuscript "Downregulation of mitochondrial biogenesis by virus infection triggers antiviral responses by cyclic GMP-AMP synthase" for consideration at PLOS Pathogens. As with all papers reviewed by the journal, your manuscript was reviewed by members of the editorial board and by several independent reviewers. The reviewers appreciated the attention to an important topic. Based on the reviews, we are likely to accept this manuscript for publication, providing that you modify the manuscript according to the review recommendations.

Both reviewers felt you manuscript greatly improved, please consider the minor concerns still left. Most of them concern a better explication of the data presented (or how they were analyzed).

Sincerely,

Matthias Johannes Schnell, PhD

Associate Editor

PLOS Pathogens

Benhur Lee

Section Editor

PLOS Pathogens

Kasturi Haldar

Editor-in-Chief

PLOS Pathogens

orcid.org/0000-0001-5065-158X

Michael Malim

Editor-in-Chief

PLOS Pathogens

orcid.org/0000-0002-7699-2064

Reviewer Comments (if any, and for reference):

Reviewer's Responses to Questions

**Part I - Summary**

Reviewer #1: The authors have provided substantial new data to bolster their conclusions and the manuscript is improved. However, there are some remaining issues related to the original critique that were not resolved and should be addressed:

Reviewer #2: The revised manuscript by Sato et al. is greatly improved and addresses all of my previous concerns. The authors clearly show that measles virus, as well as RSV and MVA, induce the liberation of mitochondrial DNA at later stages during infection, which in turn causes activation of the cGAS/STING pathway. The in vivo experiments with cGAS-/- mice suggest that this pathway indeed contributes to the total antiviral response initiated by the host. The manuscript is well written and the data support the authors' conclusions.

It is still unclear why and how this downregulation of mitochondrial biogenesis and release of mitochonrial DNA occurs during MeV infection, but this will clearly require extensive research beyond the scope of this manuscript.

**Part II – Major Issues: Key Experiments Required for Acceptance**

Reviewer #1: 1. In the PGC-1a knock-down cells only mitochondrial potential was measured. Mitochondrial membrane potential can change without any change in mitochondrial biogenesis/abundance. To better correlate this phenotype to the MeV phenotype, the same measures of mitochondrial abundance should be performed in both experiments.

2. The authors have still not provided evidence that RNA sensing is occurring before mtDNA sensing as stated by their "two-step model." The ISRE reporter does does not distinguish between RNA and DNA innate immune sensing. Also, when the mitochondrial changes are occurring (early or late) is key to this model. Thus, temporal ordering of RNA signaling, mitochondrial changes and DNA sensing is still needed to bolster the authors conclusions with regard to the two-step model.

3. The western blots showing purity of cytoplasmic fraction from all experiments should be shown even they all look the same.

Reviewer #2: none

**Part III – Minor Issues: Editorial and Data Presentation Modifications**

Reviewer #1: Figure S2a "nucleus" is misspelled

Reviewer #2: I have only one issue the authors should consider:

The qPCR data in Fig. 4C and S4 is presented as "Gene XYZ / MeV N". However, as shown in Fig. 4B, MeV N greatly differs between wild type and cGAS-/- mice (at least some of them). Therefore, the calculated lower values for the Gene XYZ / MeV N ratios in cGAS-/- mice may mostly be due to the increased MeV N expression. It would be more accurate to show all gene expression relative to the housekeeping gene (according to M&M 28S rRNA). In this case we could better assess whether there is less IFNb or other ISG expression observed in cGAS-/- mice.

PLOS authors have the option to publish the peer review history of their article (what does this mean?). If published, this will include your full peer review and any attached files.

Reviewer #1: No

Reviewer #2: No

Figure Files:

Data Requirements:

Reproducibility:

References:

---

## [Editor Report · Decision Letter 2]

27 Jul 2021

Dear Prof. Kai,

We are pleased to inform you that your manuscript 'Downregulation of mitochondrial biogenesis by virus infection triggers antiviral responses by cyclic GMP-AMP synthase' has been provisionally accepted for publication in PLOS Pathogens.

Best regards,

Matthias Johannes Schnell, PhD

Associate Editor

PLOS Pathogens

Benhur Lee

Section Editor

PLOS Pathogens

Kasturi Haldar

Editor-in-Chief

PLOS Pathogens

orcid.org/0000-0001-5065-158X

Michael Malim

Editor-in-Chief

PLOS Pathogens

orcid.org/0000-0002-7699-2064
---

## [Editor Report · Acceptance letter]

14 Sep 2021

Dear Prof. Kai,

We are delighted to inform you that your manuscript, "Downregulation of mitochondrial biogenesis by virus infection triggers antiviral responses by cyclic GMP-AMP synthase," has been formally accepted for publication in PLOS Pathogens.

Best regards,

Kasturi Haldar

Editor-in-Chief

PLOS Pathogens

orcid.org/0000-0001-5065-158X

Michael Malim

Editor-in-Chief

PLOS Pathogens

orcid.org/0000-0002-7699-2064